# Safety of Combined Targeted and Helixor^®^ *Viscum album* L. Therapy in Breast and Gynecological Cancer Patients, a Real-World Data Study

**DOI:** 10.3390/ijerph20032565

**Published:** 2023-01-31

**Authors:** Friedemann Schad, Anja Thronicke

**Affiliations:** 1Network Oncology, Research Institute Havelhöhe, Kladower Damm 221, 14089 Berlin, Germany; 2Oncological Center, Department of Interdisciplinary Oncology and Palliative Care, Hospital Gemeinschaftskrankenhaus Havelhöhe, Kladower Damm 221, 14089 Berlin, Germany

**Keywords:** safety analysis, *Viscum album* L., Helixor^®^ VA therapy, targeted therapy, monoclonal antibody therapy, immune checkpoint inhibitors, tyrosine kinase inhibitors, PARP inhibitors, CDK 4/6 inhibitors, breast cancer, ovarian cancer, endometrium cancer

## Abstract

Background: Newer personalized medicines including targeted therapies such as PARP inhibitors and CDK 4/6 inhibitors have been shown to improve the survival of breast and gynaecological cancer patients. However, efficacy outcomes may be ham5pered by treatment discontinuation due to targeted therapy-related adverse drug reactions or resistance. Studies have suggested that add-on mistletoe (*Viscum album* L., VA) improves the quality of life and ameliorates the cytotoxic side effects of standard oncological therapy in cancer patients. The primary objective of this real-world data study was to determine the safety profile of targeted therapy in combination with add-on Helixor^®^ VA therapy compared to targeted therapy alone in breast and gynecological cancer patients. Methods: The present study is a real-world data observational cohort study utilizing demographic and treatment data from the accredited national Network Oncology (NO) registry. The study has received ethics approval. The safety profile of targeted therapies with or without Helixor^®^ VA therapy and safety—associated variables were evaluated by univariate and adjusted multivariable regression analyses. Results: All stages of breast and gynecological cancer patients (*n* = 242) were on average 54.5 ± 14.2 years old. One hundred and sixty patients (66.1%) were in the control (CTRL, targeted therapy) and 82 patients (33.9%) were in the combinational (COMB, targeted plus Helixor^®^ VA therapy) group. The addition of Helixor^®^ VA did not hamper the safety profile (χ^2^ = 0.107, *p*-value = 0.99) of targeted therapy. Furthermore, no adverse events and a trend towards an improved targeted therapy adherence were observed in the COMB group. Conclusions: The present study is the first of its kind showing the applicability of Helixor^®^ VA in combination with targeted therapies. The results indicate that add-on Helixor^®^ VA does not negatively alter the safety profile of targeted therapies in breast and gynaecological cancer patients.

## 1. Introduction

Targeted therapy as the foundation for precision medicine consists of small molecules or monoclonal antibodies that aim to inhibit or induce proteins controlling the proliferation, growth and mobility of cancer cells [1]. Testing cancer cells for specific biomarkers can help to choose the right targeted therapy, either a monoclonal antibody (mAB), a tyrosine kinase inhibitor (TKI), a poly(ADP-ribose)polymer—inhibitor (PARP inhibitor or PARPi) or a cyclin-dependent kinase (CDK) 4/6 inhibitor (CDKi). PARPi, e.g., olaparib, rucaparib, niraparib, talazoparib, and velaparib, target breast or ovarian cancer cells overexpressing PARP and help to mediate the inhibition of homologous recombination deficiencies in cells [2]. CDKis, e.g., palbociclib, ribociclib, and abemaciclib, inhibit the cell cycle progression of cancer cells in patients with advanced or metastatic hormone receptor (HR)—positive and human epidermal growth factor receptor 2 (EGF-R)—negative cancer [3]. Some drawbacks might be the side effects of targeted therapy, which can hamper the patient’s quality of life [1], resistance against targeted therapy, and unexpected treatment responses due to off-target effects [2]. Most of the side effects disappear with the treatment’s end and this may be the reason why some of the targeted therapies will not continuously be taken by the patients or not continuously applied leading to treatment discontinuations and hampering of the efficacy of the treatment. Systematic reviews have shown that less than 60% of European Medicines Agency (EMA) or Federal Drug Administration (FDA) (accelerated) oncology drug approvals comprising mABs and TKIs between 2009 and 2013 (EMA) or between 2008 and 2012 (FDA) were effective in improving the overall survival and health related quality of life [4,5,6]. *Viscum album* L. (VA, European white-berry mistletoe) applied to anti-cancer standard oncological therapy in order to improve the health-related quality of life of cancer patients [7,8,9] revealed a sound safety profile when combined with targeted therapies such as monoclonal antibodies or immune checkpoint inhibitors [10,11]. Furthermore, it was shown that the addition of VA to targeted therapy was associated with the significant reduction of adverse event (AE) rates in mAB treated cancer patients [10], halved AE rates in Nivolumab-treated advanced or metastasized lung cancer patients [12] and helped to maintain standard oncological therapy in mAB, TKI or immune checkpoint inhibitor (ICI)-treated cancer patients [13]. We therefore hypothesized that the application of Helixor^®^ VA would not hamper the safety profile of applied targeted therapy in breast and gynecological cancer patients.

## 2. Materials and Methods

### 2.1. Study Design

The safety of targeted therapy with or without concomitant Helixor^®^ VA extracts was examined in a real-world data study utilizing data from the oncological registry, Network Oncology. Patients received Helixor^®^ VA therapy subcutaneously according to the summary of product characteristics (SmPC). The off-label intravenous application of Helixor^®^ VA therapy was performed in individual cases. The rationale for VA application in the patients of the current study was the improvement of health-related quality of life and self-regulation by meliorating cancer and therapy related symptoms. VA was administered at the discretion of the physician. The primary outcome of this real-world data observational cohort study was to investigate the occurrence of AEs during targeted therapy treatment with and without Helixor^®^ VA to assess the AE rate in breast and gynecological cancer. Thus, the research goal was the evaluation of the association of the outcome AE frequency with the add-on application of Helixor^®^ VA to targeted therapy.

The secondary outcome was the explorative analysis of factors that were associated with the risk of experiencing an AE or treatment adjustment of targeted therapy.

### 2.2. Description of Study Participants

Breast and gynecological cancer patients that have been registered in the Network Oncology (NO), an accredited German clinical registry [14], were enrolled in the study until January 2022. The following patients were included: patients who were 18 years or older and of both genders, and patients who gave written consent and who received targeted therapy with or without concomitant Helixor^®^ VA therapy. The study was approved by the ethics committee of the Medical Association Berlin (Eth-27/10).

### 2.3. Data Source and Assessment

Demographic data and information about the diagnosis and treatment were extracted from the NO registry. Aside from the targeted therapy and Helixor^®^ VA therapy, further information about the applied chemotherapy, radiation and surgery was analyzed. AEs were designated according to the ICH guidelines topic E2A [15] and defined as “*any untoward medical occurrence in a patient or clinical investigation subject administered a pharmaceutical product and which does not necessarily have to have a causal relationship with this treatment*”. In terms of severity, AEs were also evaluated according to the Common Terminology Criteria for AEs (CTCAE) version 5 and designated as serious or non-serious according to the ICH guidelines. AEs were classified as preferred terms according to the Medical Dictionary for Regulatory Activities (MedDRA^®^) Version 24.1 and grouped by system organ classes (SOC).

### 2.4. Statistical Methods

We described the continuous variables as the median with the interquartile range (IQR), while the categorical variables are indicated as frequencies and percentages. Furthermore, the data distributions were inspected graphically using box plot and histogram images and arithmetically examined for skewness. A stepwise backward variable selection with the Akaike information criterion was performed for the consideration of the parameters within regression models. All *p*-values <0.05 were considered as significant. All statistical analyses were performed using the R software (Version 4.2.2, R core team, Vienna, Austria), a language and environment for statistical computing. For both groups, baseline characteristics and treatment regimens were compared using the unpaired two-sided Student’s *t*-test for independent samples when the data were normally distributed. For non-normally distributed ordinal data the Mann-Whitney U test was applied. For a comparison of the categorial variables, chi-square analysis was performed. All tests were performed two-sided. Univariate two-sided Fisher’s exact test or chisquare statistical analysis were performed to detect differences in the AEs, dose reduction, or treatment discontinuation rates between the groups. Adjusted multivariable regression analysis with the binary outcome (experienced AE/treatment adjustment—yes/no) was performed to identify associated factors within the study group. We adjusted for the following parameters: age (in years), tumor origin (breast cancer, ovarian cancer, other cancer including endometrium cancer, tubal cancer, vulva carcinoma, and cervical carcinoma), targeted therapy (mAB, CDK 4/6 inhibitor, TKI, PARPi, ICI), add-on Helixor^®^ VA therapy (yes/no), UICC stage (early I–II/advanced III–IV), surgery (yes/no), chemotherapy (yes/no), and radiation (yes/no). If applicable, Brier scores as comparisons of predicted risks with observed outcomes at the individual level where outcome values were either 0 or 1 were indicated [16]. Furthermore, Nagelkerke’s R2 values as percentages of variation in the outcome explained by the predictors in the model, were indicated, if applicable.

## 3. Results

### 3.1. Baseline Characteristics

In total, 242 patients were treated, and of these 160 patients (66.1%) underwent targeted therapy without add-on Helixor^®^ VA therapy (control group, CTRL) and 82 patients (33.9%) underwent targeted and add-on Helixor^®^ VA therapy (combinational group, COMB) until January 2022 (see flowchart, Figure 1).

The mean age of the total cohort was 54.6 ± 14.2 years. The most prevalent cancers were breast cancer (87.6%) and ovarian cancer (10.3%). Nine patients (2.1%) had other cancer, see Table 1. The cancer entities and targeted therapies were balanced between the groups. The highest proportion of patients in the COMB group were patients with breast cancer (91.5%) followed by patients with ovarian cancer (7.3%). The same order with respect to proportions was observed in the CTRL group. About 17.5% of all-stage breast cancer patients and 62% of UICC stage IV breast cancer patients in the present study were hormone receptor positive and HER2-negative, data not shown. About 58.5% of the breast cancer patients were HER2-positive, data not shown.

### 3.2. Oncological Treatment

Cancer-related surgery was performed in 208 patients (86%) and radiation was perfomed in 140 patients (57.9%) with almost balanced proportions of patients in both treatment groups, respectively (CTRL_surgery_ 87.5% vs. COMB_surgery_ 82.9%; CTRL_radiation_ 55% vs. COMB_radiation_ 63.4%), see Table 2. Chemotherapy (CTx) was applied to 229 patients (94.6%), with balanced proportions in both groups, see Table 2. The median duration of chemotherapy was 147 days (interquartile range—IQR: 92–360 days). Regarding targeted therapy, monoclonal antibodies were applied most often (79.8%), followed by CDKis (10.7%), ICIs (5.4%), TKIs (2.5%) and PARPis (1.7%), see Table 3.

### 3.3. Characterization of Targeted Therapy

Table 3 characterizes the various targeted therapies applied to the two different groups. The proportions of applied targeted therapy types were balanced between both groups. The median duration of the targeted therapy was 295.5 days (IQR 108.5–476.5 days). Twenty-six patients (10.7%) received CDK 4/6 inhibitors with the highest proportion of patients receiving palbociclib, see Table 3. Two hundred six (84.1%) patients received monoclonal antibodies with trastuzumabn and a combination of pertuzumab/trastuzumab was the most applied mAB followed by bevacizumab. Atezolizumab was the most applied immune checkpoint inhibitor (4.5%) in these patients, followed by pembrolizumab. Four (1.7%) patients received PARP-inhibitors with olaparib being more often applied (1.2%) than niraparib (0.4%). The latter was not applied in the COMB group. Finally, yet importantly, six patients received TKI with lapatinib (1.7%) being more often applied than erlotinib, nintedanib and pazopanib (all 0.4%). The latter three were not applied in the COMB group. Even though some of the mABs, TKIs and PARP inhibitors were not applied in the COMB group, probably due to the smaller sample size than the CTRL group, the spectrum of targeted therapy was balanced between both groups as shown by the non-significant difference in Table 3.

Regarding the add-on VA treatment in the COMB group, Helixor^®^ A (68.3%) and Helixor^®^ M (25.6%) were the most frequently applied Helixor^®^ VA extracts in the study group, data not shown. Thus, the most applied combinations of Helixor^®^ A, M and P were with monoclonal antibodies (81.7%). Furthermore, Helixor^®^ A and Helixor^®^ M were also combined with CDKis (11%), see Table 4, mainly abemaciclib, palbociclib and ribociclib. In addition, both Helixor^®^ remedies (only in breast cancer patients) were combined with the immune checkpoint inhibitors (6.1%) such as atezolizumab and pembrolizumab. Lapatinib was the only TKI (2.4%) that was combined with Helixor^®^ VA in breast cancer patients. Figure 2 represents the specific treatment pattern of targeted therapy in the COMB group, where patients received Helixor^®^ VA therapy in addition to the targeted therapy. Here, the mAB combination of pertuzumab and trastuzumab (39%) was the most applied targeted therapy followed by the monocolonal antibodies bevacizumab (14.6%) and denusomab (4.9%), the immune checkpoint inhibitor atezolizumab (4.9%), and the CDKis palbociclib (4.9%) and abemaciclib (3.7%). Univariate analysis revealed that there were no significant correlations between additional Helixor^®^ VA treatment and age (χ^2^ = 55.5, *p* = 0.53), additional Helixor^®^ VA treatment and surgery (χ^2^ = 0.01, *p* = 0.919) and add-on Helixor^®^ VA treatment and tumor type (χ^2^ = 2.28, *p* = 0.13), data not shown. A clinically meaningful significant association was found between add-on Helixor^®^ VA treatment and increasing UICC stage (χ^2^ = 12.37, *p* = 0.00044).

### 3.4. AEs Related to Targeted and Combinational Treatment

The total AE frequency was 4.1% (regarding the number of AEs per total patient number), with 10 AEs in 242 patients, see Table 4. With respect to the treatment groups, 10 AEs (6.3%) were observed in the CTRL, and no AEs weres observed in patients of the COMB group. The AE frequencies (χ^2^ = 0.107, *p*-value = 0.99) did not significantly differ between both groups. No serious AEs or serious adverse reactions (ICH) [15] were documented for the total study cohort. No deaths from toxic effects of the studied drugs were reported.

The most often reported AE in the total study cohort was nausea, see Table 4. In terms of system organ class (SOC) classification, most AEs were ‘general disorders and administration site conditions’ (*n* = 5; 2.1% for the total group, 3.1% for the CTRL group), followed by ‘gastrointestinal disorders’ (*n* = 4; 1.7% for the total group, 2.5% for the CTRL group), and, skin and subcutaneous tissue disorder’ (*n* = 1; 0.4% for the total group, 0.6% for the CTRL group), see Table 4 and Figure 3.

We investigated in the next step whether the targeted therapy was continuously applied or whether any differences in the disruption or dose reduction of the therapy were observed between both groups. Table 5 shows the difference between the groups in the proportions of patients who experienced a targeted therapy-related AE, a dose reduction, or discontinuation of targeted therapy.

We found that the proportion of patients experiencing any of these three events was smaller in the group receiving additional Helixor^®^ VA (COMB) and the Pearson’s chi-square analysis indicated a negative association between the parameters (χ^2^ = 0.02, *p* = 0.88), but was not statistically significant. Regarding the therapy adjustment, we observed five (3.1%) disruptions of targeted treatment in the CRTL group with four discontinuations of trastuzumab (mAB) therapy and one of denusomab (mAB). Furthermore, we observed five dose reductions (3.1%) in the CRTL group, with one dose reduction of pertuzumab (mAB), one of trastuzumab (mAB), one of niraparib (PARP inhibitor), one of pazopanib (TKI), and one dose reduction and disruption of ribociclib (CDKi) therapy, see Table 5. In the COMB group, the following information was documented: one dose discontinuation (1.2%) of pertuzumab and three (3.7%) dose reductions (two of pertuzumab and one of the pertuzumab/trastuzumab therapy), see Table 5.

### 3.5. Factors Associated with Occurrence of AE, Treatment Discontinuation or Dose Disruption in Targeted Therapy

Multivariable logistic regression analysis adjusting for demographic (age), treatment-related (surgery, chemotherapy, radiation, targeted therapy, Helixor^®^ VA therapy) and tumour stage-related (UICC) variables revealed a negative correlation between AE/treatment adaption and Helixor^®^ VA application, see Figure 3. Here we found lower odds representing a 56% reduced (but not significant) probability of AE/dose reduction/treatment disruption of the targeted therapy when Helixor^®^ VA therapy was added compared to no addition of Helixor^®^ VA therapy, see Figure 3. These results are not significant; however, the Nagelkerke’s R2 value of 0.20 indicates a medium size effect according to Cohen. Thus, no significant associations between the safety profile of the targeted treatment and the Helixor^®^ VA treatment were seen. Interestingly, radiation and or the therapy with PARP-inhibitors significantly increased the probability of an AE/treatment adaption, see Figure 4. Conversely, a former chemotherapy was associated with a significant reduction in AEs and treatment adaption of the targeted therapy. Age, tumor stage, surgery, and the addition of mABs or TKIs did not haveany significant effect on the AE/treatment adaption of the targeted therapy.

## 4. Discussion

In the present observational real-world data study we evaluated the safety profile of targeted therapy in combination with add-on Helixor^®^ VA therapy in breast and gynecological cancer patients. Our findings show that additional Helixor^®^ VA therapy does not impair the safety profile of the targeted therapy in breast and gynecological cancer patients.

Regarding the differences in the baseline characteristics of exposure groups, we found no significant misbalances in age, tumor stage, tumor type or oncological treatment. A slightly higher (but not significant) proportion of patients receiving radiation therapy in the group receiving additional Helixor^®^ VA therapy might be correlated with a slightly higher (but not significant) percentage of early tumors in this group [17,18]. While the proportions of breast and ovarian cancer patients seem to be balanced between the control and in the combinational group, other cancer types such as cervix cancer were under-represented. Regarding systemic oncological treatment, the high proportion of patients receiving trastuzumab in both groups was correlated with the high proportion of patients with breast cancer overexpressing the HER2/neu protein [19]. Another targeted therapy group is that with CDKi, mainly represented in our study by palbociclib, which has been approved for hormone receptor positive, HER2/neu-negative, locally advanced or metastasized breast cancer in combination with an aromatase inhibitor [20]. Palbociclib was applied in the present study by 7.4% of breast cancer patients who were in the same range of available stage IV HR+ and HER2– breast cancer patients. The third targeted therapy group documented in our study was the ICI group, which in most cases, was atezolizumab, which has been approved for the therapy of triple-negative (HR–/HER2–) metastasized or locally advanced unresectable breast cancer with PD-L1 expression ≥1% [21]. Concerning the fourth targeted group in our study, with PARP inhibitors, olaparib has been approved for patients with advanced ovarian cancer with BRCA mutations in the fourth line, while niraparib has been approved as a maintenance therapy for high-grade serous non-mucinous epithelial ovarian cancer. Ovarian cancer was rare in our study according to the proportionally small application rate of the orphan drugs olaparib and niraparib. The fifth targeted group in our study was the group with TKIs, mostly represented by lapatinib, which is used in breast cancer overexpressing HER2/neu [22]. Thus, our real-world data study mirrors the current application situation of the five mentioned targeted groups including mABs, ICIs, TKIs, PARPis and CDKis in breast and gynecological cancer patients. Furthermore, it is the first of its kind to show the applicability of Helixor^®^ VA applications in addition to these targeted therapy groups.

As we could not see any significant differences between both groups regarding targeted therapy—related discontinuation, dose reduction or AE rates, we conclude that the combinational therapy with Helixor^®^ VA did not impair the safety profile of the targeted therapy. Interestingly, compared to a 3% rate in the control group, no AEs were observed in the combinational group indicating a possible trend towards lower AE rates here. We also observed this trend for treatment discontinuations with lower rates in the targeted therapy plus Helixor^®^ VA group. The differences were not significant though, probably due to the small sample size. Our results are in line with three other observational studies indicating the safety of the combined treatment of VA extracts with monoclonal antibodies [10], immune checkpoint inhibitors [11] and targeted therapy including mABs, ICIs, and TKIs in oncological patients [13].

In the present study no significant changes in the safety profile of targeted therapy due to add-on Helixor^®^ VA were observed. Nevertheless, former clinical studies have shown that in combination with chemotherapy, Helixor^®^ VA therapy reduces the AE rate in cancer patients and thus helped to maintain standard oncological therapy.

In a multicenter, randomized open prospective clinical trial it was shown that Helixor^®^ VA therapy that was concomitantly applied with polychemotherapy significantly reduced the AEs including fatigue and pain in 233 patients with breast, ovarian or non-small cell lung cancer [23]. The same study also revealed that the quality of life in patients receiving the combinational therapy was significantly improved [23]. In another open randomized clinical trial, tumor- and therapy-related side effects (pain, nausea/vomiting, diarrhea and appetite loss) were reduced in breast cancer patients receiving chemotherapy (CAF) plus a Helixor^®^ VA containing mistletoe therapy compared to those receiving only chemotherapy (CAF) [24]. This in turn led to fewer discontinuations of the standard therapy. A former real-world data study investigated the safety profile of mAB therapies with add-on Helixor^®^ VA therapy in oncological patients [10]. Here, the probability of an AE was five times higher in those patients that were treated with mAB therapy compared to patients treated with a combination of mAB and Helixor^®^ VA therapy [10]. The possible reason we did not see such changes may be, that breast and gynecological cancer patients could represent a cancer class with better-tolerated targeted therapy regimes and could therefore have a potentially lower risk of therapy discontinuations compared to other cancer patients who were not evaluated in this study. We could show this discrepancy in a former study where we found that breast cancer patients compared to patients with e.g., gastrointestinal or respiratory oncological diseases had a 94% reduced risk of the discontinuation of standard oncological therapy [13].

The non-randomized character of this study limits our results. However, the compared groups were balanced, thus reducing the risk of comparing heterogeneous patient groups regarding the tumor type, disease stage, and oncological treatment. In addition, biases were reduced by multivariable logistic regression methods in the safety analyses addressing potential confounders. The frequency of AEs seemed to be lower than the AE frequency reported in clinical studies [25,26,27,28], indicating a possible underreporting of AEs in our study. We assume that this may be due to the documentation and spontaneous reporting biases being in line with a systematic review [29]. Therefore, our findings may need to be cautiously interpreted. Nevertheless, our study represents the real-world application situation of targeted therapy with or without add-on Helixor^®^ VA therapy.

## 5. Conclusions

The present observational real-world data study reveals a first insight into the safety aspects of concomitant targeted and Helixor^®^ VA treatment in breast and gynecological cancer patients. The results indicate that add-on Helixor^®^ VA did not negatively alter the safety profiles of any of the five targeted therapy groups investigated. Furthermore, no adverse events and a trend towards improved targeted therapy adherence were observed in the COMB group. In addition, the adjusted multivariable regression analysis revealed a trend towards a reduction of disadvantageous events, including AEs, dose reductions, and treatment discontinuations. Further clinical studies with larger cohorts and the inclusion of additional cancer entities are being initiated.

## Figures and Tables

**Figure 1 ijerph-20-02565-f001:**
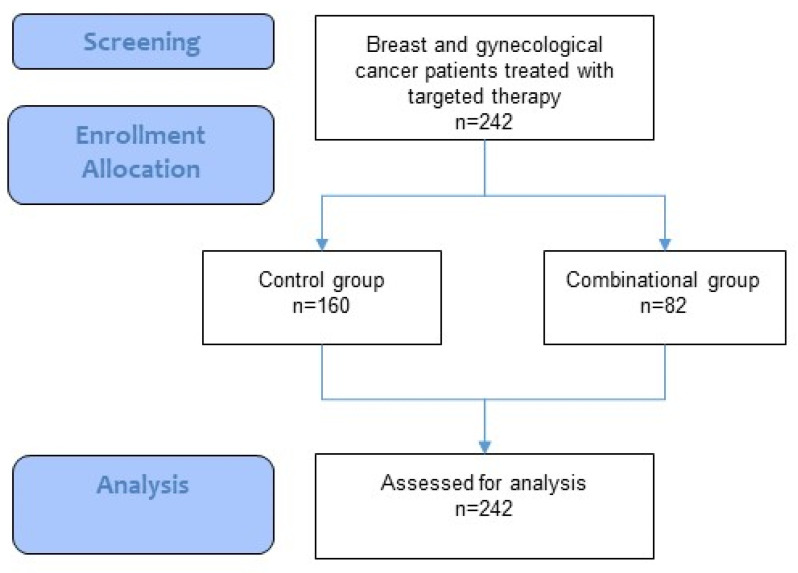
Flowchart of the study.

**Figure 2 ijerph-20-02565-f002:**
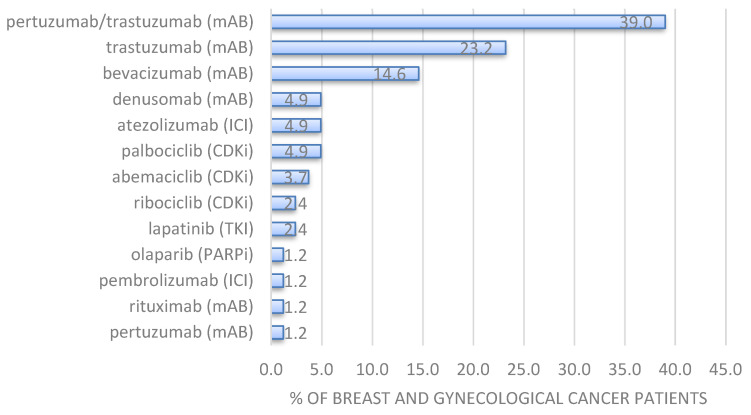
Proportion of cancer patients receiving various targeted therapies in combination with Helixor^®^ VA therapy (COMB group, *n* = 82). CDK 4/6 inhibitors (CDKi), immune checkpoint inhibitors (ICI), monoclonal antibodies (mAB), PARP-inhibitors (PARPi), TKI-inhibitors (TKI).

**Figure 3 ijerph-20-02565-f003:**
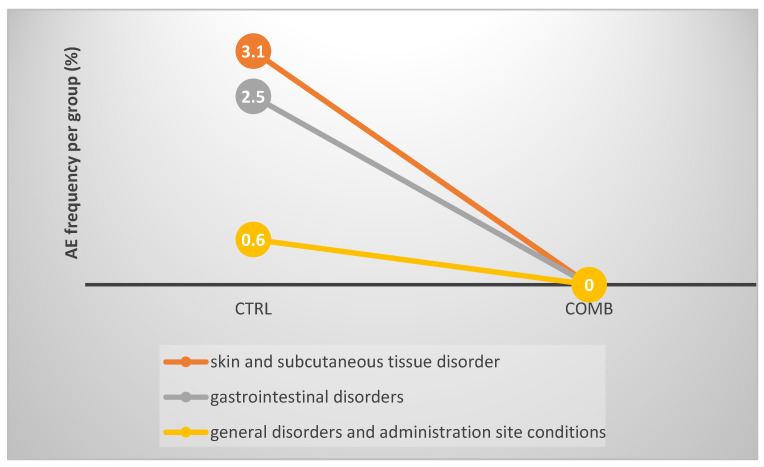
AE frequency during targeted therapy treatment per group in % illustrated as system organ class categories. Left, % AE control group, right, % AE combinational group.

**Figure 4 ijerph-20-02565-f004:**
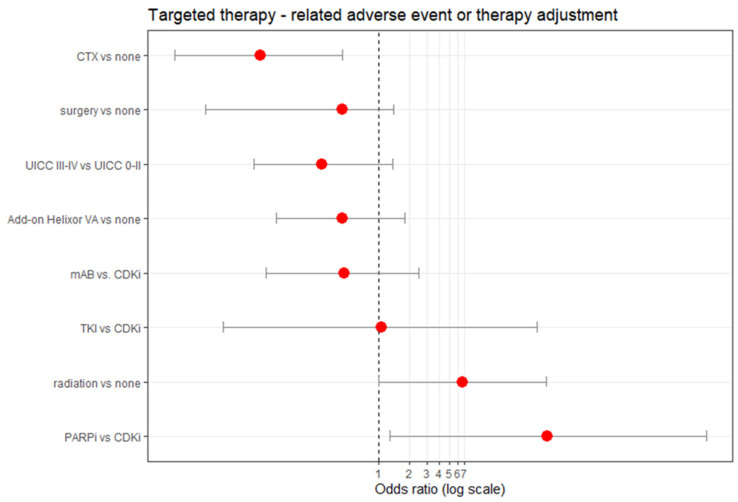
Association factors for a targeted therapy—related adverse event or therapy adjustment (dose reduction, treatment disruption). Multivariable logistic regression (*n* = 222) adjusting for age, the UICC tumor stage and add-on Helixor^®^ VA treatment, as well as surgery, chemotherapy (CTX), radiation and the respective type of targeted therapy. Red dots indicate the odds ratio (OR). CDKi, CDK 4/6-inhibitors; ICI, immune checkpoint inhibitors; mAB, monoclonal antibody; PARPi, PARP-inhibitor; TKI, TKI-inhibitors; AE, adverse event; Brier Score, 0.046, Nagelkerke_R2, 0.2145.

**Table 1 ijerph-20-02565-t001:** Characteristics of patients.

Patient Characteristics	Total Cohort *n* = 242	CTRL *n* = 160	COMB *n* = 82	Significance *p*-Value
Age at first diagnosis, years, mean (SD)	54.5 (14.2)	54.7 (14.4)	54.3 (13.7)	0.84
Cancer entity, *n* (%)				1.13
Breast, *n* (%)	212 (87.6)	136 (85)	75 (91.5)	
Ovarian, *n* (%)	25 (10.3)	19 (11.9)	6 (7.3)	
Other ^1)^, *n* (%)	5 (2.1)	5 (3.1)	0 (0)	
UICC stage				0.08
0	1 (0.4)	1 (0.6)	0	
I	49 (20.2)	26 (16.3)	23 (28.0)	
II	86 (35.5)	55 (34.4)	31 (37.8)	
III	49 (20.2)	39 (24.4)	10 (12.2)	
IV	44 (18.2)	29 (18.1)	15 (18.3)	
NA	13 (5.4)	10 (6.3)	82 (3.7)	

Characteristics of the patients included in the study, total cohort and respective treatment groups; IQR, interquartile range, ^1)^ other cancer entities include endometrium, tubal, vulva and cervical cancer; *n*, number; %, percent; UICC, Union for International Cancer Control.

**Table 2 ijerph-20-02565-t002:** Characteristics of oncological therapy.

	Total Cohort, *n* = 242	CTRL *n* = 160	COMB *n* = 82	Significance *p*-Value
Surgery				0.91
yes	208 (86.0)	140 (87.5)	68 (82.9)	
no	27 (11.2)	19 (11.9)	8 (9.8)	
NA	7 (2.9)	1 (0.6)	6 (7.3)	
Radiation				0.08
yes	140 (57.9)	88 (55.0)	52 (63.4)	
no	95 (39.3)	71 (44.4)	24 (29.3)	
NA	7 (2.9)	1 (0.6)	6 (7.3)	
CTx				0.51
yes	229 (94.6)	153 (95.6)	76 (92.7)	
no	13 (5.4)	7 (4.4)	6 (7.3)	

CTx, chemotherapy; *n*, number; %, percent.

**Table 3 ijerph-20-02565-t003:** Characterization of targeted therapy.

Patient Characteristics	Total Cohort *n* = 242	CTRL *n* = 160	COMB *n* = 82	Significance *p*-Value
CDKi, *n* (%)	26 (10.7)	17 (10.6)	9 (11.0)	1.0
abemaciclib, *n* (%)	4 (1.7)	1 (0.6)	3 (3.7)	
palbociclib, *n* (%)	18 (7.4)	14 (8.8)	4 (4.9)	
ribociclib, *n* (%)	5 (2.1)	3 (1.9)	2 (2.4)	
mAB, *n* (%)	206 (84.1)	136 (85.0)	70 (85.4)	1.0
bevacizumab, *n* (%)	47 (19.4)	35 (2.19)	12 (14.6)	
denusomab, *n* (%)	8 (3.3)	4 (2.5)	4 (4.9)	
glembatumumab, *n* (%)	1 (0.4)	1 (0.6)	0	
pertuzumab, *n* (%)	2 (0.8)	1 (0.6)	1 (1.2)	
pertuzumab/trastuzumab, *n* (%)	62 (25.6)	30 (18.8)	32 (39.0)	
rituximab, *n* (%)	2 (0.8)	1 (0.6)	1 (1.2)	
trastuzumab, *n* (%)	74 (30.6)	55 (34.4)	19 (23.2)	
trastuzumab-emtasin, *n* (%)	3 (1.2)	3 (1.9)	0	
ICI, *n* (%)	13 (5.4)	8 (5)	5 (6.1)	1.0
atezolizumab, *n* (%)	11 (4.5)	7 (4.4)	4 (4.9)	
pembrolizumab, *n* (%)	2 (0.8)	1 (0.6)	1 (1.2)	
PARPi, *n* (%)	4 (1.7)	3 (1.9)	1 (1.2)	1.0
niraparib, *n* (%)	1 (0.4)	1 (0.6)	0	
olaparib, *n* (%)	3 (1.2)	2 (1.3)	1 (1.2)	
TKI, *n* (%)	6 (2.5)	4 (2.5)	2 (2.4)	1.0
erlotinib, *n* (%)	1 (0.4)	1 (0.6)	0	
lapatinib, *n* (%)	4 (1.7)	2 (1.3)	2 (2.4)	
nintedanib, *n* (%)	1 (0.4)	1 (0.6)	0	
pazopanib, *n* (%)	1 (0.4)	1 (0.6)	0	

Targeted therapy including CDK 4/6 inhibitors (CDKi), immune checkpoint inhibitors (ICI), monoclonal antibodies (mAB), PARP-inhibitors (PARPi), TKI-inhibitors (TKI) and combinations of them; *n*, number; %, percent. Numbers of patients in the various therapy groups may not sum up to 100 percent as some patients have received a combination of targeted therapies.

**Table 4 ijerph-20-02565-t004:** Adverse events per treatment group.

System Organ Class	Adverse Event (AE)	Total *n* = 242	CTRL *n* = 160	COMB *n* = 82
gastrointestinal disorders	nausea	2	a ^1)^, d ^4)^	-
	appetite loss	1	d ^4)^	-
	vomiting	1	d ^4)^	-
general disorders and administration site conditions	pain	1	c ^3)^*	-
	temperature elevated	1	b ^2)^*	-
	fatigue	1	e ^5)^	-
	neck stiffness	1	e ^5)^	-
	impaired vision	1	e ^5)^	-
skin and subcutaneous tissue disorder	erythema	1	e ^5)^	-
Total number of AEs		10	10	-
Total number of patients experiencing AE		5	5	0
AE per patient frequency (AE events divided by number of all patients, n (%) ^§)^		4.13% ^§)^	6.25%	0
Patient with AE frequency (patient experiencing an AE divided by number of all patients) ^§§)^		2.07% ^§§)^	3.13%	0

Adverse events per treatment group classified as MedDRA (MedDRA Version 24.1) preferred terms and grouped by system organ class; each letter a-e indicate a different patient, same letters indicate the same patient; ^1)^ pertuzumab; ^2)^ trastuzumab; ^3)^ trastuzumab/pertuzumab, ^4)^ niraparib; ^5)^ pazopanib, * treatment was discontinued; AE, adverse event; ^§)^ comparison of AE frequency COMB group vs. CTRL group: χ^2^ = 0.107, *p*-value = 0.99; ^§§)^ comparison of AE frequency COMB group vs. CTRL group: χ^2^ = 1.3, *p*-value = 0.25.

**Table 5 ijerph-20-02565-t005:** Discontinuation/dose reduction or AE due to targeted therapy.

	Total Cohort *n* = 242	CTRL *n* = 160	COMB *n* = 82	Significance *p*-Value
AE due to targeted treatment, *n* (%)	5 (2.1)	5 (3.1)	0	0.254
Disruption of targeted treatment, *n* (%)	6 (2.9)	5 (3.1)	1 (1.2)	0.642
Dose reduction of targeted treatment, *n* (%)	8 (3.3)	5 (3.1)	3 (3.7)	1.0
Any event (AE or therapy adjustment ^§)^, *n* (%)	14 (5.8)	10 (6.3)	3 (4.9)	0.586

Information on the continuation of targeted therapy; ^§)^ comparison of COMB group vs. CTRL group: (χ^2^ = 0.020, *p* = 0.88).

## Data Availability

All relevant data are within the manuscript.

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
