# Peer review of "Safety of Combined Targeted and Helixor® Viscum album L. Therapy in Breast and Gynecological Cancer Patients, a Real-World Data Study"

_ijerph, 2023, doi:10.3390/ijerph20032565_

Round 1

Reviewer 1 Report

The paper is very well written and work is scientifically sound. Methodology is proper by all means. Kindly go through the manuscript in order to make some minor grammatical and English language corrections. Also reduce the plagiarism.

Thank you

Author Response

Response to Reviewer 1 Comments

Point 1: The paper is very well written and work is scientifically sound. Methodology is proper by all means. Kindly go through the manuscript in order to make some minor grammatical and English language corrections. Also reduce the plagiarism.

Response 1: Thank you for your review of our manuscript and your evaluation that it , is very well written and work is scientifically sound,. According to your suggestion, we have made grammatical and English language corrections and have reduced the plagiarism.

Reviewer 2 Report

Thank you for the opportunity to review your interesting article.

I suggest few minor corrections:

1. The study design is not clearly mentioned in the "Methods" chapter, as well as in the "abstract". Please mention the research type.

2. The research goals are not clear enough. I recommend to present the research goals in term of associations between exposure X and outcome Y

Author Response

Response to Reviewer 2 Comments

Point 1: Thank you for the opportunity to review your interesting article.

I suggest few minor corrections:

  1. The study design is not clearly mentioned in the "Methods" chapter, as well as in the "abstract". Please mention the research type.

Response 1: Thank you for your review of our manuscript. According to your suggestion, we have indicated the study design in the ,methods, and ,abstract, chapter. The study is a real-world data observational cohort study. The PECO criteria ,population, exposures and outcomes,  are all indicated in the method chapter ,study design,.

Point 2:

  1. The research goals are not clear enough. I recommend to present the research goals in term of associations between exposure X and outcome Y

Response 2: Thank you for this important point. We have now presented the research goal as follows in line 85: “Hereby, the research goal was the evaluation of the association of the outcome AE frequency with the additional application of Helixor® VA to targeted therapy”.

Reviewer 3 Report

Schad and Thronicke intended to evaluate the safety profile of targeted therapy in combination with add-on Helixor® VA therapy in breast and gynecological cancer patients. In this work, the authors showing the applicability of Helixor® VA in combination with targeted therapies, demonstrate that add-on Helixor® VA does not negatively alter the safety profile of targeted therapies in breast and gynaecological cancer patients.

The work is well written, but there are several minor issues that can be resolved, listed below.

11. Please, in line 45 the authors also insert the full name of PARP.

22.  In lines 50-51, please insert the acronym of epidermal growth factor receptor 2.

33.  In line 57, please insert the full name of EMA and FDA.

44. In lines 64-65, the authors write: “it was shown that the addition of VA to targeted therapy was associated with the significant reduction of AE rates in mAB treated cancer patients”. What do the authors mean by AE? Adverse effects? Please specify what is meant by AE.

55.  In line 67, please indicate the full name of ICI.

66.  Format the table 1 caption in the correct form.

77. The text contains references that are not listed at the end of the work, for example in line 81, 98, 128, 234. Please review the list of references.

Author Response

Response to Reviewer 3 Comments

Point 1: Schad and Thronicke intended to evaluate the safety profile of targeted therapy in combination with add-on Helixor® VA therapy in breast and gynecological cancer patients. In this work, the authors showing the applicability of Helixor® VA in combination with targeted therapies, demonstrate that add-on Helixor® VA does not negatively alter the safety profile of targeted therapies in breast and gynaecological cancer patients.

The work is well written, but there are several minor issues that can be resolved, listed below.

  1. Please, in line 45 the authors also insert the full name of PARP.

Response 1: Thank you for your review and your evaluation that the work is ,well written,. According to your first suggestion we have inserted the full name of PARP in line 46.

Point 2:

  1. In lines 50-51, please insert the acronym of epidermal growth factor receptor 2.

Response 2: Thank you, we have inserted the acronym in line 52. 

Point 3:

  1. In line 57, please insert the full name of EMA and FDA.

Response 3: We have inserted the full names of EMA and FDA, thank you for this point. See line 59.

Point 4:

  1. In lines 64-65, the authors write: “it was shown that the addition of VA to targeted therapy was associated with the significant reduction of AE rates in mAB treated cancer patients”. What do the authors mean by AE? Adverse effects? Please specify what is meant by AE.

Response 4: Thank you for this important point. We have meant ,adverse events, and have specified AE, see line 67.

Point 5:

  1. In line 67, please indicate the full name of ICI.

Response 1: Thank you, we have now indicated the full name of ICI, immune checkpoint inhibitor, see line 70.

Point 6:

  1. Format the table 1 caption in the correct form.

Response 6: We have formatted the table 1 caption in the correct form, line 161-162.

Point 7:

  1. The text contains references that are not listed at the end of the work, for example in line 81, 98, 128, 234. Please review the list of references.

Response 7: Thank you for this important point. We have reviewed the list of references and have included the references from line 93, 104, 135 and 235 in the reference list.
